# The 22q11.2 Low Copy Repeats

**DOI:** 10.3390/genes13112101

**Published:** 2022-11-11

**Authors:** Lisanne Vervoort, Joris Robert Vermeesch

**Affiliations:** Center for Human Genetics, KU Leuven, 3000 Leuven, Belgium

**Keywords:** low copy repeats, genomic disorders, 22q11.2 deletion syndrome

## Abstract

LCR22s are among the most complex loci in the human genome and are susceptible to nonallelic homologous recombination. This can lead to a variety of genomic disorders, including deletions, duplications, and translocations, of which the 22q11.2 deletion syndrome is the most common in humans. Interrogating these phenomena is difficult due to the high complexity of the LCR22s and the inaccurate representation of the LCRs across different reference genomes. Optical mapping techniques, which provide long-range chromosomal maps, could be used to unravel the complex duplicon structure. These techniques have already uncovered the hypervariability of the LCR22-A haplotype in the human population. Although optical LCR22 mapping is a major step forward, long-read sequencing approaches will be essential to reach nucleotide resolution of the LCR22s and map the crossover sites. Accurate maps and sequences are needed to pinpoint potential predisposing alleles and, most importantly, allow for genotype–phenotype studies exploring the role of the LCR22s in health and disease. In addition, this research might provide a paradigm for the study of other rare genomic disorders.

## 1. Introduction

The acrocentric chromosome 22 is the second shortest human autosome with a length of 50 Mb of which 15 Mb belong to the p-arm. Together with the Y-chromosome, this chromosome has the highest density of low copy repeats (LCRs). LCRs are larger than 1 kb blocks of DNA that are duplicated across several inter- and intrachromosomal loci in the genome [1,2]. Due to their high sequence identity, they are subject to meiotic misalignment and recombination, a mechanism known as nonallelic homologous recombination (NAHR). Rearrangements between the LCRs on chromosome 22 (LCR22s) cause the 22q11.2 deletion syndrome (22q11.2DS), the most common genomic disorder in humans [3].

LCRs constitute 6.6% of the human genome [4], and several genes within the LCRs have played an important role in human adaptation [5]. LCRs are frequently removed in standard analysis pipelines. Due to the multiple mapping options, reads originating from LCRs are frequently wrongly assigned, creating errors and gaps in reference assemblies.

The study of the 22q11.2 deletion syndrome is limited by the lack of a proper genomic reference sequence in the 22q11.2 LCRs. The complexity of the LCR22s is characterized by a patchwork of subunits, present in multiple and variable copy numbers that share a high percentage of sequence identity. Consequently, genome sequences cannot be assembled properly, and the reference genomes have an inaccurate representation of the LCR22s.

## 2. Instability of the LCR22s in the Human Genome

The 22q11.2 locus is considered one of the most unstable loci in the human genome due to the presence of a cluster of eight LCR22s. They are commonly denoted as LCR22-A to -H and act as ideal substrates for nonallelic homologous recombination (NAHR), resulting in several chromosomal conditions (Figure 1).

### 2.1. Deletions and Duplications

Rearrangements among the four proximal LCR22s, -A, -B, -C, and -D, cause the 22q11.2DS and the reciprocal duplication syndrome [6] (Figure 1). Medical manifestations of the 22q11.2DS include congenital heart defects, immunodeficiency, and palatal anomalies. However, the syndrome is characterized by a high interpatient phenotypic variability. In 85% of individuals with the 22q11.2DS, NAHR occurs between LCR22-A and –D, the two largest LCR22s, resulting in a 3 Mb deletion (Figure 1). This region contains 46 protein-coding genes and less well characterized transcripts within the LCR22s. In 90–95% of patients, the deletion is de novo [6,7].

Nested deletions are defined as deletions where NAHR involves either LCR22-B or -C [6]. Of 22q11.2DS patients, 5% carry the 1.5 Mb LCR22-A/B deletion. The 2 Mb LCR22-A/C deletion, the 1.5 Mb LCR22-B/D deletion, and the 1 Mb LCR22-C/D deletion account for 2%, 4%, and 1% of patients, respectively (Figure 1) [7]. Given that these deletions cause a milder phenotype, they are more frequently inherited. Deletions encompassing the distal LCR22s (LCR22-E until -H) are less frequent, and the phenotypes differ from the traditional 22q11.2DS [8]. These deletions are referred to as the distal 22q11.2 deletion syndrome (MIM: 611867) [9]. Rearrangements where one of the breakpoints is located in the unique sequence between the LCR22s have been observed as well and are termed atypical deletions [8]. The prevalence of these nonstandard (nested, distal, and atypical) deletions is likely underestimated since they are associated with milder phenotypes and are, therefore, harder to diagnose. In addition, not all genetic tests are able to detect these uncommon deletions (for example, standard 22q11.2del FISH using probes TUPLE or N25 are located in the unique sequence between LCR22-A and -B).

Reciprocal to the standard 22q11.2DS, the 22q11.2 duplication syndrome (MIM: 608363) has been described [10]. Symptoms include velopharyngeal insufficiency, behavioral deficits, and dysmorphic features, and overlap with the 22q11.2DS.

### 2.2. Translocations

LCR22s are essential for the creation of several nonrandom constitutional translocations. Recurrent translocations include t(8;22)(q24.1;q11.2), t(11;22)(q23;q11.2), and t(17;22)(q11.2;q11.2), whereas t(1;22)(p21.1;q11.2) and t(4;22)(q35.1;q11.2) belong to the nonrecurrent class [11]. Translocations can directly affect the function of a gene; for example, t(17;22) disrupts *NF1* on chromosome 17, leading to neurofibromatosis type 1 [12].

The best known non-Robertsonian translocation is t(11;22)(q23;q11.2). Since there is no gain or loss of DNA, carriers are phenotypically normal, except for reproductive problems [13]. They can have phenotypically normal offspring when the child inherits the two normal chromosomes or the two translocated chromosomes, as a result of balanced 2:2 segregation during meiosis. Gametes and embryos following unbalanced 2:2 segregation can be conceived as well. Such conceptions lead to spontaneous miscarriage, explaining the high rates among carrier parents [14]. However, the offspring can inherit the derivative chromosome from the translocation-carrying parent in addition to the normal nontranslocation partner chromosomes. This mechanism is known as 3:1 meiotic malsegregation [13] and can lead to clinical symptoms since the copy number is changed. For example, carriers of the t(11;22)(q23;q11.2) translocation have a 10% risk of having a child with the supernumerary der(22)t(11;22) syndrome. This syndrome, also known as Emanuel syndrome (MIM: 609029), is caused by the presence of a partial trisomy of the 11q23 and the 22q11 locus. Clinical features of the syndrome include ear pits, micrognathia, heart malformations, and cleft palate [15]. The breakpoints of these constitutional 22q11.2 translocations all cluster in LCR22-B (Figure 1), described by Emanuel in 2008 as the most rearrangement-prone site of the human genome [16].

Several nonconstitutional translocations involving 22q were said to play a role in carcinogenesis as well. The chromosomal abnormality resulting from a translocation between chromosome 9q34 (ABL locus) and chromosome 22q11 (BCR locus) is called the Philadelphia chromosome. The *BCR-ABL1* fusion gene encodes a hybrid protein important in the disease pathway of several forms of leukemia [17]. In addition, the t(8;22) translocation is a variant of the t(8;14) translocation associated with Burkitt’s lymphoma [18], and Ewing sarcoma (MIM: 612219) is caused by the t(11;22)(q24;q12) translocation fusing the *FLI1* and *EWS* genes on chromosomes 11 and 22, respectively [19]. These oncogenic translocations arise via mitotic rearrangement and are only present in the tumor or affected tissue.

### 2.3. Cat Eye Syndrome

A more complex constitutional LCR22-mediated rare disorder is cat eye syndrome (MIM: 115470), characterized by iris coloboma, anal atresia, and ear malformations [20]. The karyotype marks the presence of a small supernumerary dicentric, bisatellited chromosome representing an inv dup(22) (q11.2) [20,21]. This structure leads to a tetrasomy of the chromosome 22 p-arm and part of the 22q11.2 locus (from the centromere to the distal breakpoint). The breakpoints are mapped to LCR22-A and/or -D, and different cytogenetic cat eye syndrome types exist based on the specific LCR22 involvement (Figure 1) [20]: type I and IIb marker chromosomes are symmetrical with both breakpoints clustering in LCR22-A or -D, respectively. Asymmetric cat eye syndrome marker chromosomes characterized by one LCR22-A and one LCR22-D breakpoint are classified as type IIa. The supernumerary chromosome is predicted to be created via interchromosomal misalignment or intrachromosomal recombination, leading to an inversion with subsequent homologue crossover [21].

Hence, the presence of LCR sequence on chromosome 22 makes the locus prone to several types of genomic rearrangements, leading to disease and disease predisposition.

## 3. LCR22 Assembly and Representation in the Reference Genome

In 1999, chromosome 22 was the first finished human chromosome sequence [22]. This sequence covered the euchromatic part and consisted of 12 contigs spanning 33.4 Mb. Four of the 11 gaps were located in the LCR22-associated 22q11.2 locus [22]. In 2008, 8 of the 11 gaps were closed by combining different sequencing techniques [23]. However, the 22q11.2 locus still harbored two of the remaining gaps. The inability to differentiate between two alleles on homologous chromosomes or two (or more) paralogues on the same chromosome (or a combination) is the main cause for these failures [1] The length of the LCR22 modules surpasses the length of second generation sequencing reads, and misalignments can lead to false structural variant calls.

As a consequence, comparisons between the different reference genomes showed several inconsistencies at the level of the LCR22s. A first organizational overview of the LCR22 structure was provided even before the release of the first human reference genome assembly [24] (Figure 2). Based on sequencing data from a bacterial artificial chromosome (BAC), a P1-derived artificial chromosome, and cosmid insert clones, four LCR22s were differentiated, each represented as a mosaic patchwork of different modules. These modules are shared between one or more LCR22s via duplication and, therefore, have high sequence similarity (97–98%). For the two largest blocks, LCR22-A and -D, the lengths were estimated at 350 and 250 kb, respectively. They share long stretches of identical duplicon composition, except for the LCR22-A-specific proximal USP18 (yellow) and distal DGCR6/PRODH module (red) and the LCR22-D-specific PI4KA module (mustard) (Figure 2). The smaller LCR22-B and -C blocks are composed of modules that are present in LCR22-A and/or LCR22-D, but they do not share similar blocks between them, suggesting that rearrangements between the smaller LCR22s are not possible. Hence, the duplicated module structure of the LCR22s provides ideal elements for NAHR.

Despite the successful structural representation of LCR22 by Shaikh et al. (2000), difficulties were encountered when assembling the LCR22s during the generation of the human reference genome [25]. In hg19 (GRCh37, released February 2009), the LCR22s were delineated (Table 1) with a gap of 100 kb present in LCR22-B (Figure 2). Surprisingly, LCR22-A was identified as the second smallest LCR22. In the reference genome GRCh38, released in December 2013, relative LCR22 sizes were comparable to the first representation [24] with major changes compared with hg19 (Figure 2, Table 1). First, a 100 kb module, including the genes RIMBP3 and TMEM191B and pseudogene PI4KAP1, was shifted from LCR22-B to LCR22-A. Second, no gaps were present anymore in LCR22-B, but three new gaps appeared in LCR22-A with lengths of 100 (most proximal), 50, and 50 kb (most distal). Due to the variability and inconsistencies between the different genome assemblies, constructing the exact composition and sequence of the LCR22s is still an ongoing challenge.

Despite their importance in disease predisposition, the extreme duplication nature of the LCR22s causes difficulties in standard mapping pipelines, assembly errors, and incomplete representation in the reference genomes. Global structural information is missing in the short reads and standard long reads, hindering a correct assembly of this structurally complex locus. There is a need for de novo, nonreference based approaches to correctly compile the LCR22 haplotype.

## 4. Entering the Era of Optical Mapping

Long-range information over the repeat is necessary to decipher the exact structural composition of the LCR22s. Applications to achieve this long-range structural conformation include optical mapping techniques, such as fiber-FISH (fluorescent in situ hybridization) and Bionano optical mapping. These tools enable the de novo assembly of structurally complex loci with a resolution of 5–10 kb [26]. In Bionano optical mapping, fluorescently labeled DNA fibers are scanned, and the de novo assembly is based on the labeling pattern of the reads (Figure 3). Due to the lengths of these fibers (N50 > 150 kb in general), large deletions, duplications, inversions, and translocations can be visualized. Using this technique, LCR haplotypes and additional structural variants were discovered in the 7q11.23, 15q13.3, and 16p12.2 locus [27]. The method is not targeted, and therefore, structural variants can be inferred at a whole-genome scale [28]. In fiber-FISH, one specific locus of the genome is scrutinized using fluorescently labeled probes targeting the region of interest (Figure 3). In brief, long DNA molecules (>300 kb) are stretched onto a glass surface and hybridized with probes [29]. This approach has proven to be successful for structural variation detection [30,31].

Optical mapping techniques were able to elucidate the LCR22 architecture for the first time [32]. To the same end, fiber-FISH was performed on the haploid cell line CHM1 and the diploid cell line GM12878. LCR22 haplotypes were de novo assembled by tiling fibers based on matching colors and distances between the probes [32]. The smaller LCR22s-B and -C were identical to the composition observed in the reference genome hg38. In both cell lines, the three haplotypes of LCR22-D were identical to each other, but differed from the hg38 reference by an inversion distal in LCR22-D. Three different haplotypes were compiled for LCR22-A, with lengths between 0.65 and 1.20 Mb and substantially different from the reference genome hg38. Bionano optical mapping was performed on these two cell lines as an orthogonal method to validate the results [32].

To evaluate whether the variation between these three different LCR22-A haplotypes in the three investigated alleles was a coincidence, a population-scale study was performed in 187 individuals and uncovered the presence of 6 LCR22-D variants and 26 complete LCR22-A configurations [32]. The LCR22 haplotyping of 57 diploid genomes (114 alleles of the parents of patients with 22q11.2DS) validated the presence of 5 LCR22-D configurations and extended the LCR22-A catalogue to 32 unique haplotypes [33]. Through optical mapping, visualization of the complex LCR22 repeat structure and LCR22-A variation was possible for the first time.

The expansion and variation of LCR22-A was examined in an evolutionary context, by mapping the LCR22 alleles of great apes (chimpanzee, bonobo, gorilla, orangutan) and one macaque (New World monkey) [34]. The optical mapping data were able to correct misassemblies and close gaps in the existing corresponding reference genomes. In addition, the haplotypes of the investigated samples from the *Pan* genus (five chimpanzees and one bonobo) showed no variation and represented a small, ‘minimal’ haplotype. Therefore, the expansion and variation observed in the human population can be considered human specific [34].

## 5. The Hunt for the Rearrangement Sites

Pinpointing the crossover sites at nucleotide resolution will give us insights into the mechanisms behind these deletions, the genes involved in the rearrangement, and eventually enable us to conduct phenotype–genotype correlation studies. Unfortunately, nucleotide level identification is complicated by the duplication nature.

Once again, optical mapping techniques have allowed for initial attempts at identifying rearrangements at the LCR22 subunit/duplicon level. In Demaerel et al. (2019), the seven LCR22-A/D recombinations all mapped to a 160 kb module (Figure 4). Pastor et al. (2020) mapped 30 LCR22-A/D deletion trios, and the same 160 kb locus or smaller loci were involved. All family maps demonstrate the association of the rearrangement with the FAM230 locus (Figure 4), raising the potential for this locus to be the preferred 22q11.2DS recombination hotspot.

A subclass of 22q11.2 deletions is atypical rearrangements, where one of the breakpoints resides in an LCR22 and the second breakpoint is located in an unique sequence surrounding the LCR22s [24,37,38,39,40,41,42,43,44,45], offering an opportunity to map the rearrangement sites via the unique sequence. In the International 22q11.2 Brain Behavior Consortium [46], six individuals were identified with atypical deletions. A combination of whole-genome sequencing, fiber-FISH, patient-specific PCR, and targeted long-read sequencing was used to investigate these deletions. They were all located at different loci within the LCR22. Based on the presence of microhomology or polyA insertion at the breakpoints, they likely originated via replication-based mechanisms [47].

The distal LCR22 blocks (LCR22-E until -H) and LCR22-B and -C are substantially smaller compared with LCR22-A and -D, facilitating targeted approaches. Shaikh et al. (2007) used long-range PCR with paralogous-specific primers to amplify and subsequently sequence the breakpoints of two distal 22q11.2 deletions. The breakpoint cluster region (*BCR*) module was identified as the recombination locus in both an LCR22-D/E and an LCR22-E/F deletion [35]. The breakpoints of the constitutional 22q11.2 translocations all cluster in LCR22-B, described by Emanuel in 2008 as the most rearrangement-prone site of the human genome [16]. By scrutinizing these rearrangements, palindromic AT-rich repeats (PATRRs) were identified at the breakpoint locus in LCR22-B as well as in the involved partner chromosome locus, generating instability and an opportunity for rearrangement via secondary structure formation [11]. Although an important role in the generation of translocations, the contribution of palindrome-mediated pathways to deletions and duplications in the 22q11.2 locus cannot yet be confirmed [16].

To solve the complex LCR22-A/D rearrangement, Guo et al. (2016) charted shared and paralogous sequence polymorphisms between LCR22-A and -D by sequencing BACs containing (parts of) the LCR22s [35]. Based on this variation map, whole-genome sequencing data of two LCR22-A/D patients and their parents were screened for uniquely mapping read pairs within these LCR22s. The results suggested that the BCR module was involved in the recombination mechanism of two recurrent LCR22-A/D deletions (Guo et al. 2016). These studies propose that the BCR module is the hotspot for 22q11.2 chromosomal rearrangements (Figure 4).

The breakpoints of the 22q11.2 rearrangements coincide in the LCR22s, providing evidence for the presence of specific elements involved in nonhomologous rearrangement mechanisms. Both the FAM230 [33] and BCR module [35,36] were suggested as 22q11.2 recombination hotspot. Interestingly, neither of them are present in any of the four proximal LCR22s (Figure 4), and therefore, at least two recombination spots will be involved in the creation of the nested deletions involving LCR22-B and -C. It remains to be seen whether other rearrangement sites are involved and whether there is variability observed at the nucleotide level. In order to paint the complete ‘recombination landscape’ at the 22q11.2 locus, more deletions should be mapped.

## 6. The Future: Ultralong Read Sequencing Approaches?

Long-range structural information at the base-pair level will be needed to map the LCR22s and pinpoint the nucleotide rearrangement positions. To reach this goal, long-read technology from Oxford Nanopore holds promise. Average read lengths are dependent on the input material and library preparation method, but reads have been reported to be as long as 2.27 Mb [48]. Error rates are lower than 5% and are continuously improving [49]. ONT sequencing of 3622 Icelanders has shown that the technique is able to accurately characterize structural variants at a population scale [50]. A revolution in CNV research is underway using high-throughput Nanopore sequencing.

Combinations of second- and third-generation sequencing tools and optical mapping techniques are the gold standard for whole-genome de novo assembly [51,52]. The telomere-to-telomere consortium generated the first complete assembly of a human genome, including unresolved loci, such as ribosomal rRNA, centromeric satellites, and segmental duplications. Using ultralong read Nanopore (39X coverage and N50–70 kb) and SMRT PacBio (70X coverage) data, de novo assembly of the homozygous complete hydatidiform mole CHM13 genome was achieved. This initial draft was polished by (shorter) Nanopore and PacBio, linked-read Illumina (10X Genomics), and Bionano optical mapping data. In April 2022, the first complete human genome [4] was assembled. Using this CHM13 haploid cell line with only a single LCR22-A allele, all LCR22s were completely sequenced. Projecting this sequence onto the duplicon structures of the optical mapping patterns described by Demaerel et al. (2019), this assembly represents a common LCR22-A haplotype. Unfortunately, T2T-CHM13 LCR22-A lacks the SD22-3 duplicon. The absence of this duplicon may stymie future analyses using T2T-CHM13 as a reference genome. Nevertheless, the release of this new reference genome makes it possible to compare novel sequences against a more accurate, reliable reference genome and is a major step forward.

## 7. Remaining Questions and Future Perspectives

Major progress in 22q11.2DS research has been made with the elucidation of the LCR22 structure via optical mapping techniques. New questions may now be probed in the future. What is the extent of this LCR22-A variability? What is the effect of this variability at the transcriptome level? Is there variability in the rearrangement locus, and is this genotype variability associated with the observed phenotypic variability? Are specific haplotype lengths or structures a risk factor for the rearrangements?

First, it will be essential to further catalogue the LCR22 variation at large scale. In addition, new resources and databases must be constructed to collect and visualize the human (LCR22) structural variation, as begun by the Human Pangenome Reference Consortium [53]. They will establish a database of 350 phased genomes, generating a total of 700 haplotypes [53]. The development of bioinformatic tools to solve and construct complex structural variants in a high-throughput way will accelerate future 22q11.2 research, including crossover site identification on a larger scale. This initiative will expand the catalogue of gross 22q11.2 structural variation at the duplicon and sequence level, thus generating a valuable control population dataset.

Second, the consequences of this discovered LCR22 variability must be examined. LCR22-specific structural variation could cause (I) gene-dosage effects in the LCR22-specific transcripts based on copy number variation or new fusion transcripts associated with specific rearrangements and (II) effects on gene expression in LCR22 flanking genes, and (III) exert a genome-wide impact. Of particular interest are the genes located within the LCR22-A sequence (FAM230, RIMBP3, PI4KAP, etc.). Given that an accurate gapless reference genome has only recently become available, genes and transcripts within the LCR22s are poorly characterized. Both regular and haplotype-resolved Hi-C analysis mapping long-range chromosomal interactions on 22q11.2DS patients demonstrated that LCR22-A and -D act as topologically associated domains [54]. Thus, a nucleotide difference of over 1.75 Mb between the smallest and largest LCR22-A haplotype observed will likely affect topologically associated domain boundaries and may exert an effect at the transcriptional level.

With these major advances, genotype–phenotype association studies are now on the horizon and can be performed to associate haplotype configurations or transcriptomic alterations with a phenotypic feature. These studies may finally begin to interrogate a long-held hypothesis in the field. Since LCR genes in the genome are known to play a role in human neuronal development [5], the neuropsychiatric phenotypic consequences will be of particular interest.

## Figures and Tables

**Figure 1 genes-13-02101-f001:**
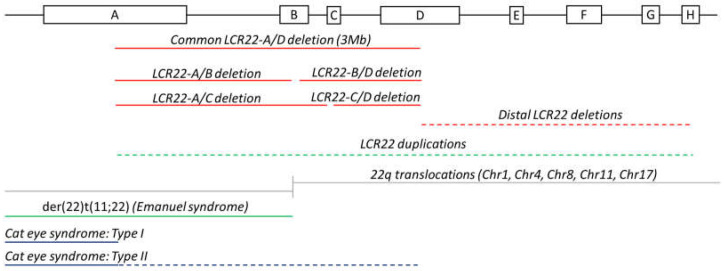
Rearrangements between low copy repeats on chromosome 22. Organization of LCR22s-A until -H. Schematic overview of the rearrangements observed between LCR22s with dotted lines indicating the options of crossovers between different blocks. The red lines indicate deletions. The green lines represent duplications. The grey lines show the rearrangement locus in the recurrent translocations involving chromosome 22. The dark blue lines show the triplication of a fragment in cat eye syndrome. The dashed lines cover different-sized CNVs.

**Figure 2 genes-13-02101-f002:**
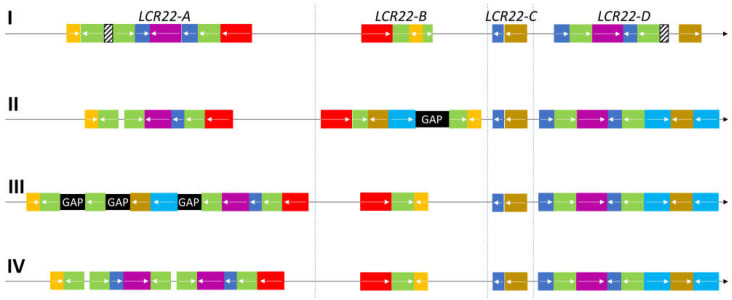
Organization of LCR22-duplicated modules in several references. (I) First description of the LCR22 structure in Shaikh et al. (2000). These modules serve as ‘essential duplicons’ in the comparison and composition of the LCR22s in the other reference genomes (II–IV). The yellow duplicon contains USP18 (or a paralogue). The red duplicon encompasses (paralogues of) the genes PRODH and DGCR6. The PI4KA gene or a paralogue is present in the mustard-colored blocks. The cyan duplicon always covers a RIMBP3 paralogue, the blue duplicon a BCR paralogue, and the magenta duplicon a GGT paralogue. A FAM230 paralogue can be found in the green duplicon, as well as a palindromic AT-rich repeat. (II) LCR22 organization in GRCh37/hg19. (III) LCR22 organization in GRCh38/hg38. (IV) LCR22 organization, for the first time without gaps, in the most recent reference genome (T2T-CHM13).

**Figure 3 genes-13-02101-f003:**
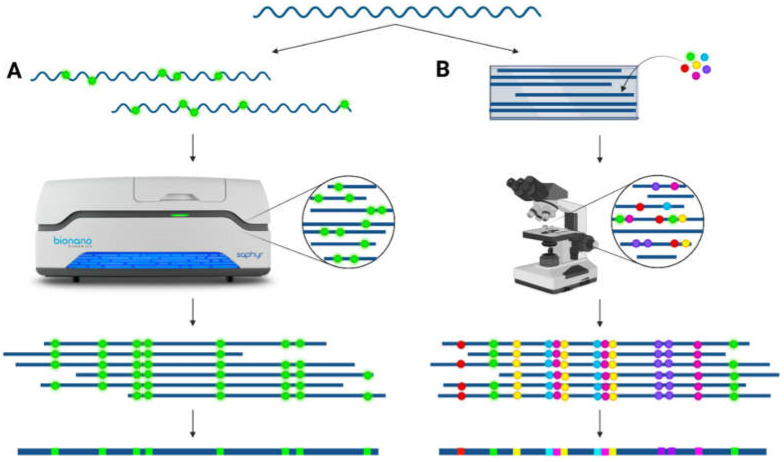
De novo assembly via optical mapping techniques. (**A**) Bionano optical mapping: ultralong (>100 kb) DNA fibers are fluorescently labeled on specific locations across the whole genome and subsequently linearized and scanned. These images are converted into molecules to visualize the labeling pattern across the DNA strand. By overlapping identical patterns, a consensus pattern can be created. (**B**) Fiber-FISH: ultralong (>300 kb) DNA fibers are stretched onto slides and hybridized with a combination of different fluorescently colored probes targeting the locus of interest. The slide is scanned using a microscope, allowing the detection of the different colors via scanning at different excitation levels. A consensus of the region of interest can be compiled based on overlapping colors and distances between the probes.

**Figure 4 genes-13-02101-f004:**
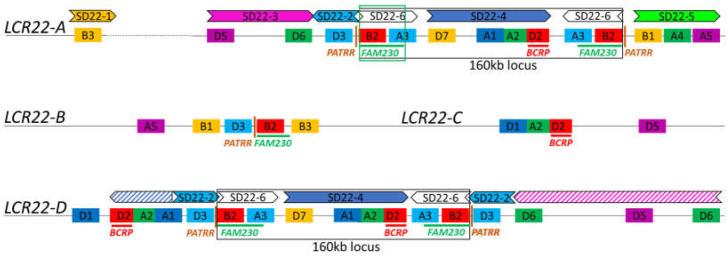
Recombination loci on the LCR22 map. The four proximal LCR22s are depicted schematically based on the fiber-FISH probe composition from Demaerel et al. (2019). LCR22-A is represented as the smallest haplotype including all possible SD22 duplicons. The dotted line reflects the variability in size, copy number, and composition. The SD22 duplicons are illustrated above the fiber-FISH probe compositions of LCR22-A and -D. The BCR module [35,36], FAM230 [33], PATRR [16], and 160 kb locus [32] are indicated in the LCR22s. Drawings are not to scale.

**Table 1 genes-13-02101-t001:** Exact chromosomal locations of the different LCR22s in hg19, hg38, and telomere-to-telomere reference genomes.

	GRCh37/hg19	GRCh38/hg38	T2T-CHM13
LCR22-A	chr22:18,639,043-19,022,986	chr22:18,156,276-19,035,473	CP068256.2:18,828,186-19,410,796
LCR22-B	chr22:20,128,537-20,731,921	chr22:20,141,014-20,377,631	CP068256.2:20,520,047-20,781,953
LCR22-C	chr22:21,021,564-21,092,560	chr22:20,667,276-20,738,272	CP068256.2:21,075,991-21,146,982
LCR22-D	chr22:21,363,668-21,916,380	chr22:21,009,379-21,562,091	CP068256.2:21,418,153-21,975,566

## Data Availability

Not applicable.

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
