# Peer review of "The 22q11.2 Low Copy Repeats"

_genes, 2022, doi:10.3390/genes13112101_

Round 1

Reviewer 1 Report

Congratulations, an exceptional review regarding LCRs, in particular at chr-22.

Author Response

We thank the reviewers for this short but supportive review. We have carefully reviewed the grammar and performed a spell check. 

Reviewer 2 Report

genes-1959965

In this review, the author makes a comprehensive review of the 22q11.2 low copy repeats and their related genomic disorders. I commend the authors for their extensive review and compiled over many years of detailed structural studies in LCR22s and their future. In addition, the manuscript is clearly written in professional, unambiguous language. If there is a weakness, it is a minor spell check and makes the figure legend format consistent. Here are some examples:

Line 60-61, Line 67-68, line 135: Please clarify why the author highlights the cited papers in the introduction (Campbell et al. 2018, Burnside 2015).

Figure1: please clarify the difference between the solid lines and dash lines.

Line 81-81: “The translocation can directly affect a gene and his function, as 81 described for t(17;22) leading to neurofibromatosis type 1 by disrupting the NF1 gene on 82 chromosome 17”. Please revise “his” to “its”. 

Please thoroughly proofread the manuscripts to correct any grammar issues and ensure that the figure legend and citation styles are consistent.

Author Response

We thank the reviewers for their supportive review comments.

We addressed the points raised:

We performed a thorough grammar and spell check. 

Line 60-61, Line 67-68, line 135: Please clarify why the author highlights the cited papers in the introduction (Campbell et al. 2018, Burnside 2015).

There is no need to highlight those references and they have been referred to with the reference numbering system. 

Figure1: please clarify the difference between the solid lines and dash lines.

We added in the figure legend:  'The dashed lines indicate a group of different sized CNVs. 

Line 81-81: “The translocation can directly affect a gene and his function, as 81 described for t(17;22) leading to neurofibromatosis type 1 by disrupting the NF1 gene on 82 chromosome 17”. Please revise “his” to “its”. 

We corrected this.